# Use of Statins in Heart Failure with Preserved Ejection Fraction: Current Evidence and Perspectives

**DOI:** 10.3390/ijms25094958

**Published:** 2024-05-01

**Authors:** Artem Ovchinnikov, Alexandra Potekhina, Tatiana Arefieva, Anastasiia Filatova, Fail Ageev, Evgeny Belyavskiy

**Affiliations:** 1Laboratory of Myocardial Fibrosis and Heart Failure with Preserved Ejection Fraction, National Medical Research Center of Cardiology Named after Academician E.I. Chazov, Academician Chazov St., 15a, 121552 Moscow, Russia; potehina@gmail.com (A.P.); anastasia.m088@yandex.ru (A.F.); 2Department of Clinical Functional Diagnostics, A.I. Yevdokimov Moscow State University of Medicine and Dentistry, Delegatskaya St., 20, p. 1, 127473 Moscow, Russia; 3Laboratory of Cell Immunology, National Medical Research Center of Cardiology Named after Academician E.I. Chazov, Academician Chazov St., 15a, 121552 Moscow, Russia; areftan2@gmail.com; 4Faculty of Basic Medicine, Lomonosov Moscow State University, Leninskie Gory, 1, 119991 Moscow, Russia; 5Out-Patient Department, National Medical Research Center of Cardiology Named after Academician E.I. Chazov, Academician Chazov St., 15a, 121552 Moscow, Russia; ftageev@gmail.com; 6Medizinisches Versorgungszentrum des Deutsches Herzzentrum der Charite, Augustenburger Platz 1, 13353 Berlin, Germany; evgeny.belyavskiy@dhzc-charite.de

**Keywords:** statins, heart failure with preserved ejection fraction, inflammation, diastolic dysfunction, prognosis

## Abstract

Systemic inflammation and coronary microvascular endothelial dysfunction are essential pathophysiological factors in heart failure (HF) with preserved ejection fraction (HFpEF) that support the use of statins. The pleiotropic properties of statins, such as anti-inflammatory, antihypertrophic, antifibrotic, and antioxidant effects, are generally accepted and may be beneficial in HF, especially in HFpEF. Numerous observational clinical trials have consistently shown a beneficial prognostic effect of statins in patients with HFpEF, while the results of two larger trials in patients with HFrEF have been controversial. Such differences may be related to a more pronounced impact of the pleiotropic properties of statins on the pathophysiology of HFpEF and pro-inflammatory comorbidities (arterial hypertension, diabetes mellitus, obesity, chronic kidney disease) that are more common in HFpEF. This review discusses the potential mechanisms of statin action that may be beneficial for patients with HFpEF, as well as clinical trials that have evaluated the statin effects on left ventricular diastolic function and clinical outcomes in patients with HFpEF.

## 1. Introduction

Approximately half of patients with heart failure (HF) have normal (preserved) ejection fraction (HFpEF) [1]. HFpEF is becoming one of the key problems of modern healthcare. The prevalence of HFpEF is steadily increasing due to the aging population and high incidence of arterial hypertension, obesity, and diabetes mellitus. On the other hand, this increasing prevalence of HFpEF is confounded by improved diagnostic capabilities that contribute to better detectability of HFpEF. The prognosis is almost as poor as in HF with reduced ejection fraction (HFrEF), while effective treatments for HFpEF are still limited [1]. Most therapies that improve outcomes in HFrEF have not been as effective in HFpEF, which may be explained by the difference in the initial pathophysiological process in these two major HF phenotypes: cardiomyocyte death in HFrEF and microvascular myocardial inflammation in HFpEF [2].

The results of multiple clinical trials with anti-inflammatory/immunomodulatory therapy for HFrEF (with corticosteroids, methotrexate, immunoglobulin, or tumor necrosis factor [TNF] inhibitors) have been inconclusive and contradictory [3]. The lack of success of these trials is partly due to the fact that myocardial inflammation in HFrEF is detected only at advanced stages and is a result of a reactive response to severe left ventricular (LV) systolic dysfunction; while at earlier stages, LV remodeling is controlled primarily through cardiomyocyte loss [2]. In contrast, in HFpEF, remodeling is initially governed by chronic microvascular inflammation, which is supported by compelling preclinical and clinical evidence [4].

The pro-inflammatory paradigm of HFpEF was proposed in 2013 by Paulus W.J. and Tschöpe C. [2]. Most patients with HFpEF have multiple extracardiac pro-inflammatory comorbidities such as obesity, hypertension, atrial fibrillation, type 2 diabetes mellitus, coronary artery disease, chronic kidney disease, chronic obstructive pulmonary disease, or anemia [5]. These comorbidities along with advanced unhealthy aging have been postulated to trigger a sustained low-grade systemic inflammatory response with systemic endothelial dysfunction, including microvascular coronary dysfunction. Activated endothelial cells begin to express adhesion molecules on their surface [6] with subsequent myocardial infiltration with activated leukocytes, the transformation of fibroblasts into myofibroblasts, and the development of cardiac interstitial fibrosis [2,7]. Endothelial microvascular inflammation also triggers oxidative stress, reduces nitric oxide (NO) bioavailability, and impairs the NO—cyclic guanosine monophosphate (cGMP)—protein kinase G (PKG) signaling pathway, resulting in cardiomyocyte hypertrophy, altered myofilament protein phosphorylation, and increased cardiomyocyte stiffness [8]. In turn, interstitial fibrosis and stiff cardiomyocytes contribute to increased LV stiffness and LV filling pressure with the development of HFpEF.

Thus, chronic low-intensity myocardial inflammation may play a key role in the pathogenesis of HFpEF, supporting the use of anti-inflammatory therapy. In these circumstances, 3-hydroxy-3-methylglutaryl coenzyme A (HMG-CoA) reductase inhibitors (or statins) hold considerable promise [2]. In addition to their hypolipidemic effects and a high safety profile, statins stimulate NO production in endothelial cells and exhibit anti-inflammatory, antioxidant, antihypertrophic, and antifibrotic activities [9]. In this review, we consider the potential mechanisms of statin action explored in experimental studies that may be useful in HFpEF, as well as clinical trials that have evaluated the effects of statins on LV diastolic function and clinical outcomes in patients with HFpEF.

## 2. Statins: General Presentation

Statins act as competitive and reversible inhibitors of HMG-CoA reductase, the rate-limiting enzyme of cholesterol biosynthesis, and thereby significantly reduce de novo cholesterol synthesis in the liver. In response to a decrease in cholesterol synthesis, the density of receptors to low-density lipoproteins (LDL) on the surface of hepatocytes increases, which leads to an increase in the uptake of LDL and its precursors from the bloodstream. In addition, statins inhibit the synthesis of apolipoprotein B-100 and reduce the synthesis and secretion of triglyceride-rich particles [10]. Statins moderately increase the concentration of high-density lipoprotein cholesterol and have no effect on the concentration of lipoprotein (a) and the size and density of LDL [10].

Statins reduce the blood levels of atherogenic lipoproteins and thereby the risk of coronary artery disease and its complications, regardless of blood cholesterol levels [11]. It has been shown that for every mmol/L reduction in LDL cholesterol (LDL-C) levels, there is a 22% reduction in the risk of vascular events [12]. Statins stabilize existing atheromatous plaques and prevent the formation of new plaques, thereby reducing the overall coronary burden [13], which may be of great importance for HFrEF of ischemic etiology.

In addition to their hypolipidemic effects, statins exert a plethora of pleiotropic effects without a clear relationship LDL-C levels [14]. The pleiotropic properties of statins, such as an improvement of endothelial function and increase in the bioavailability of nitric oxide [15], anti-inflammatory [16], immunomodulatory and antioxidant effects [17], and regression of cardiac fibrosis and hypertrophy [18,19] are generally accepted and may be beneficial in HF population (Figure 1). The spectrum and magnitude of the pleiotropic activity of statins depend on their chemical and pharmacokinetic properties as well as on the sensitivity of cells to these drugs [20]. The pleiotropic effects of statins are thought to be mainly due to a decrease in the production of mevalonate and its numerous metabolites, such as the isoprenoid intermediates farnesylpyrophosphate (FPP) and geranyl-geranylpyrophosphate (GGPP) [14]. FPP and GGPP are involved in posttranslational modification (prenylation) of a wide range of biological molecules, primarily small GTPases, leading to an increase in their hydrophobicity and anchoring in the plasma membrane and, thereby, their activation. Small GTPases prenylated by FPPs include members of the Ras family (K-Ras, H-Ras, N-Ras, RhoE, Rheb, and RhoB); other GTPases are prenylated by GGPP (Rac1, Rac2, Ra1A, Rap1B, RhoA, RhoB, RhoC, Cdc42, and the Rab family) [21]. Small GTPases are involved in signaling pathways controlling many important cellular processes, such as cell proliferation, differentiation, apoptosis, adhesion, migration, cytokine production, and the function of the cytoskeleton [22].

One of the key substrates for activated Rho GTPases are serine/threonine Rho-associated coiled-coil-containing protein kinases (ROCKs). ROCKs are represented by two isoforms (ROCK1 and ROCK2) that have different subcellular distribution in different tissues (vascular smooth muscle cells, endothelium, fibroblasts, inflammatory cells, and cardiomyocytes), but perform similar functions, regulating many cellular processes including proliferation, motility, and cell viability [23]. The activation of ROCKs may suppress the pro-survival phosphatidylinositol 3-kinase (PI3K)/Akt (protein kinase B) pathway and induce cardiac fibrosis (via the activation of profibrotic gene expression and transdifferentiation of fibroblasts into myofibroblasts) and myocardial hypertrophy in pathologic conditions [9]. ROCK activity has been shown to be significantly increased in the leukocytes of patients with cardiovascular diseases such as metabolic syndrome, arterial hypertension, and coronary heart disease [24,25,26]. Hence, by preventing the geranylgeranylation of Rho and its activation of ROCKs, statins may inhibit their downstream effects [9].

Statins can also realize their pleiotropic effects through epigenetic modifications, binding to some proteins and modulating their activity. Among mevalonate-independent targets of statins is histone deacetylase 2 (HDAC2) [9]. The inhibition of HDAC2 activity promotes the accumulation of acetylated histone H3 and an increased expression of the cyclin-dependent protein kinase inhibitor protein p21 [27].

## 3. Microcirculatory Endothelial Dysfunction in HFpEF and Endothelial Effects of Statins

The pivotal event in the pathogenesis of HFpEF is the development of endothelial microvascular endothelial dysfunction. A hallmark of endothelial dysfunction is a decrease in NO production and its diffusion to the underlying smooth muscle myocytes/cardiomyocytes (decreased NO bioavailability) as a result of decreased expression/activation of endothelial NO synthase (eNOS) and/or eNOS uncoupling. The decrease in NO production in HFpEF is largely due to the activation of free-radical oxidation processes within endothelial cells via nicotinamide adenine dinucleotide phosphate (NADH)-oxidase [6,28], resulting in NO binding to superoxide (O_2_^−^) with the formation of cytotoxic reactive oxidant peroxynitrite (ONOO^−^) [28]. In addition, reactive oxygen species (ROS) contribute to the reduced bioavailability of tetrahydrobiopterin (BH4), a cofactor of eNOS, leading to subsequent substrate uncoupling of eNOS with the production of O_2_^−^ instead of NO (Figure 2). To date, oxidative stress is a highly evident component of the pathophysiology of HFpEF [29].

Statins induce eNOS upregulation (increase eNOS mRNA stability) and thereby increase NO bioavailability (Figure 2) [30]. This effect may be achieved through the inhibition of the Rho/ROCK pathway [31] and the activation of PI3K/protein kinase Akt pathway [31]. In endothelial cells, ROCKs inhibit the pro-survival PI3K/protein kinase Akt pathway, resulting in decreased eNOS expression and cGMP levels [32]. ROCKs also induce NADPH-oxidase, leading to the formation of ROS. In clinical trials, statins reduced ROCK activity in patients with different cardiac diseases such as atherosclerosis, heart failure, and hypertension [26,33,34] regardless of the reduction in LDL-C blood levels [34]. The inhibition of the Rho/ROCK pathway by statins also results in decreased ex-expression of miRNA-155, which suppresses eNOS expression [35].

Other mechanisms of statins to increase NO bioavailability have also been suggested, such as a downregulation in membrane protein caveolin-1 that directly binds to and inhibits eNOS [36]; an induction of kruppel-like factor-2 mRNA that is required for eNOS expression [37]; an increase in the concentration of a cofactor for eNOS function BH4 [38]; or an increased mobilization of endothelial progenitor cells [39]. The increase in NO bioavailability with statins has been associated with an enhancement of myocardial blood flow under hypoxia [40], as well as an increase in capillary density and improvement in myocardial perfusion [41].

## 4. Microcirculatory Inflammation in HFpEF and Anti-Inflammatory Effects of Statins

Patients with HFpEF usually have elevated blood levels of pro-inflammatory cytokines, which triggers systemic endothelial dysfunction and endothelial activation [4]. Activated endothelial cells express adhesion molecules such as E-selectin, P-selectin, intercellular adhesion molecule 1 (ICAM-1), and vascular cell adhesion molecule 1 (VCAM-1), which interact reversibly with the corresponding ligands on the surface of circulating monocytes (leukocyte function-associated antigen-1 [LFA-1] and integrin dimer CD11b [αM-subunit β2 integrin] Mac-1) [42], the expression and affinity of which, in turn, is enhanced under the action of endothelial chemokines. As a result of this interaction, the movement of monocytes in coronary capillaries slows down even to a complete stop. High expression of adhesion molecules (ICAM-1 and E-selectin) was detected in the coronary microcirculation of patients with HFpEF [6]. The adhesion of monocytes to endothelial cells is a prerequisite for a key step in the whole chain of the inflammatory process: the migration of monocytes from the bloodstream into the subendothelial space. This process occurs according to a concentration gradient of chemoattractants such as monocyte chemoattractant protein-1 (MCP-1) and interleukin (IL)-8, released from the stressed myocardium. Once they have migrated into the tissue, monocytes quickly transform into macrophages and start producing the main cytokine of fibrosis—transforming growth factor-β (TGF-β) (Figure 2) [6]. TGF-β triggers the transformation of fibroblasts into myofibroblasts. The latter produce collagen intensively, resulting in the progression of fibrosis and LV diastolic dysfunction (DD). In experimental models with pressure overload, (1) features of inflammation are always found in hypertrophied myocardium together with fibrosis; (2) the areas of fibrosis and inflammation usually coincide; (3) inflammation always occurs before fibrosis; and (4) the suppression of inflammation allows to prevent fibrosis as well [43]. In biopsy studies in patients with HFpEF, a significant accumulation of activated macrophages producing TGF-β and other fibroblast triggers such as galectin-3 and osteopontin was found in the myocardium, which was associated with fibroblast activation and excessive collagen deposition [7,44,45].

Among the key pleiotropic properties of statins are their well-documented anti-inflammatory effects [14]. Statins have been shown to be effective in a number of anti-inflammatory diseases: atherosclerosis [46], myocarditis [47], diabetic nephropathy [48], Alzheimer’s disease [49], bone diseases [50], and autoimmune diseases [51]. In addition, the anti-inflammatory effects of statins can be observed at all the stages of inflammation in hypertensive heart disease [52].

Statins reduce monocyte adhesion and migration through the inhibition of expression/activity of endothelial adhesion molecules (VCAM-1, E-selectin), leukocyte integrins (Mac-1, LFA-1), and chemokines (MCP-1) and their receptors; some of these effects were independent of HMG-CoA reductase inhibition (Figure 2) [53,54,55]. One potential mechanism for the anti-inflammatory action of statins is the inhibition of pro-inflammatory cytokine production with the elimination of their activating effects on nuclear factor κ-light-chain-enhancer of activated B cells (NF-kB). NF-kB is a ubiquitous transcription factor that influences the production of pro-inflammatory cytokines such as TNF-α [14] and is involved in cell proliferation and differentiation, and immune defense [56]. According to one hypothesis, the ability of statins to scavenge free oxygen radicals and stimulate nitric oxide production may contribute to the stabilization of the IkBα protein, a natural inhibitor of NF-kB [57]. The anti-inflammatory effect of statins through the inhibition of NF-kB has been convincingly demonstrated in numerous experimental and clinical studies [58,59,60,61].

Statins may exert an anti-inflammatory impact through the regulation of the Nod-like receptor family pyrin domain containing 3 (NLRP3) inflammasomes—cytosolic protein oligomeric complexes within the immune cells. NLRP3 inflammasomes detect bacterial and viral pathogens as well as the by-products of tissue damage or cellular stress, and trigger an inflammatory response via proteolytic activation of the inflammatory cytokines, interleukin-1β and -18, by caspase-1 [62]. The NLRP3 inflammasome assembly is triggered by the NF-κB pathway, which is activated by damage-associated molecular patterns (DAMPs), pathogen-associated molecular patterns (PAMPs), and endogenous cytokines. The activation signal is provided by a variety of stimuli including extracellular ATP, mitochondria-generated ROS, and particulate matter including cholesterol crystals [63]. Statins may inhibit the activation of NLRP3 in response to atherogenic stimuli by activating the pregnane X receptor [64]. Statins can also reduce the expression of toll-like receptors 2 and 4 on the surface of immune cells and thereby prevent lipopolysaccharide-induced activation of monocytes and endothelial cells [14]. A reduced expression of toll-like receptors leads to reduced activation of the NF-κB [65] and reduced NLRP3 activation [66], and ultimately causes a switch of the immune response to anti-inflammatory responses [14]. Statins also may suppress the transcription of protein components of the inflammasome through the inhibition of NF-kB [67]. Nevertheless, the cumulative effect of statins on NLRP3 inflammasome is still a matter of debate. Lipophilic statins are thought to have a stronger effect on NLRP3 inflammasome complexes than hydrophilic statins [14]. Further studies are needed to clarify the effect of statins on inflammasomes.

One of the most sensitive markers of inflammation is C-reactive protein (CRP). CRP is a positive acute-phase reactant (plasma concentrations increase in response to inflammation, in contrast to negative acute-phase proteins such as albumin, transferrin, or transthyretin, whose concentrations decrease). It is produced in hepatocytes under stimulation with IL-6, IL-1, and TNF-α [68]. CRP can also be produced locally by macrophages in foci of inflammation [69]. In addition to being a sensitive marker of inflammation, the widespread use of C-reactive protein in clinical laboratories is largely due to its ease of determination, which is another reason for its success. The relationship between CRP levels and cardiovascular risk is well established. In patients with HFpEF, CRP is associated with a greater comorbidity burden [70] and poor outcomes [71]. Numerous studies have shown the ability of statins to reduce CRP blood levels (by about 15–30%) [72,73,74], while meta-analysis has shown that lipophilic statins are more effective in decreasing high-sensitivity CRP (hsCRP) and IL-6 in patients with HF [75]. Statin suppression of CRP effects, including the effects on monocyte MCP-1, MIP-1α, and MIP-1β cytokine synthesis and cell motility, is mediated by L-mevalonate/FPP-dependent inhibition of ERK1/2 MAP kinases [76]. In the Korean Acute Heart Failure registry, CRP was a clear prognostic marker for any form of heart failure (HFrEF, HF with mildly reduced EF, or HFpEF), but only in a subgroup of patients with EF > 40% and upper tertiles of CRP (>1.14 mg/dL) who were on statins showed a better survival trend than those who were not [77].

One of the key modulators of cell metabolism, survival, and proliferation is the PI3K/Akt pathway, and the maintenance of myocardial inflammation may be mediated through enhanced activity of this pathway in immune cells, in particular macrophages [51]. The data regarding the effect of statins on the PI3K/Akt signaling pathway are ambiguous. Reduced ROCK activity may stimulate the intracellular PI3K/Akt signaling pathway and thus enhance immune cell activation. On the other hand, some data demonstrate the inhibitory effects of statins on PI3K/Akt activation; the anticipated effect depends on cell type, experiment conditions, and applied concentrations [78]. While reducing ROCK and downstream molecule activation, statins may reduce immune cell migration to the myocardium and thereby suppress inflammation, which may ultimately benefit cardiac remodeling processes [79]. Statins decrease the expression of interferon (INF)-γ-inducible major histocompatibility complex molecules class II (MHC II) on the surface of antigen-presenting cells, thereby reducing antigen presentation to CD4^+^ T lymphocytes [80]. In different modes of induction of dendritic cell maturation, statins inhibited the expression of MHC II involved in antigen presentation and of the T cell costimulatory molecules CD80/CD86 (B7-1/2), CD40, and CD83 [81,82]. Dendritic cells derived in culture from blood monocytes in the presence of statins not only failed to provide full antigen presentation to T cells but also promoted the expansion of T cells with suppressor activity [81]. Dose-dependent inhibition of CD4+ T cell proliferation accompanied by accumulation of regulatory Foxp3+ cells by statins was detected in vitro. Statins impaired spontaneous and chemokine-induced lymphocyte migration, apparently by inhibiting cellular motility [83]. Statins inhibit monocyte differentiation to macrophages and cytokine synthesis by activated macrophages at the posttranscriptional level [84].

Other anti-inflammatory mechanisms in statins include decreased TGF-β signaling in T cells, impaired T cell activation, and the induction of T regulatory cells [80]. Ex vivo, atorvastatin suppressed the expression of neutrophil chemoattractant IL-8 produced by endothelial cells [85], and simvastatin decreased chemokine production and chemokine receptor expression in human endothelial cells and macrophages [86]. Statins may directly block LFA-1 integrin [54], which is involved in leukocyte adhesion and the migration and co-stimulatory signal transduction to T cells during antigen presentation. The activation of integrins, including LFA-1, is accompanied by conformational changes leading to an increase in the affinity for their substrates. Statins may bind to LFA-1 and prevent such conformational changes, which can partly explain the anti-inflammatory activity of statins [54].

## 5. Low NO-cGMP-PKG Pathway Activity as a Major Contributor of Myocardial Dysfunction, and the Role of Statins

The main consequence of reduced NO bioavailability in HFpEF is impaired signaling through the intracellular NO-cGMP-PKG pathway (Figure 3) [87]. NO diffusion into the underlying vascular smooth muscle cells leads to microvessel dilatation. NO also inhibits the expression of adhesion molecules, reduces microvascular inflammation, regulates platelet function, and promotes the mobilization of stem cells to maintain vascular integrity. NO also penetrates adjacent cardiomyocytes, where it binds to its intracellular receptor soluble guanylate cyclase (sGC) and activates it with subsequent formation of the secondary intracellular messenger cGMP. cGMP exerts its physiological effects mainly through the activation of cGMP-dependent protein kinase type I (PKGI). PKGI directly phosphorylates a variety of proteins, exerting numerous beneficial cardiovascular effects [88].

When NO bioavailability is low, cGMP formation and PKGI activity in cardiomyocytes are reduced. The suppression of the NO-cGMP-PKG signaling network has been clearly demonstrated in animal models of HFpEF [6,89,90]. In LV biopsies, cGMP content and PKG activity were significantly lower in patients with HFpEF compared to patients with HFrEF or asymptomatic aortic stenosis [6,8], which was associated with low NO bioavailability and high nitrosative/oxidative stress [6]. ROS can directly oxidize PKGI, reducing its activity. In patients with HFpEF, higher myocardial oxidative stress-dependent activation of eNOS leading to PKGIα oxidation was revealed [91].

The NO-cGMP-PKG pathway suppresses prohypertrophic signaling cascades, including the Gq-coupled pathway via the phosphorylation of the regulator of G protein signaling (RGS) 2/4 [92], and the calcineurin/NFAT (nuclear factor of activated T-cells) pathway via the phosphorylation of the canonical transient receptor potential calcium ions (Ca^2+^) channel (TRPC) 6 [93]. At low NO-cGMP-PKG signaling pathway activity, this inhibitory effect on prohypertrophic signaling pathways may be significantly attenuated. The cGMP-PKG pathway also prevents fibrosis, presumably by suppressing TGF-β and preventing the conversion of fibroblasts to active myofibroblasts [94]. TGF-β1 is synthesized in many myocardial cells and transmits a signal to the cell nucleus via the suppressor of mothers against the decapentaplegic (SMAD) 2/3 pathway [95]. Decreased activity of the cGMP-PKG pathway in fibroblasts is associated with the stimulation of collagen synthesis and its excessive myocardial deposition [2,87]. In contrast, increased activity of the NO-cGMP-PKG pathway attenuates signaling through the SMAD2/3 pathway and prevents the development of myocardial fibrosis [96]. In addition, the transdifferentiation of fibroblasts into myofibroblasts depends on TRPC6 activity, which is inhibited by PKGI [94].

An activated NO-cGMP-PKG pathway participates in intracellular Ca^2+^ handling and contributes to normal LV relaxation. When PKG activity is low, the phosphorylation of the following proteins, involved in active relaxation, is impaired: (1) phospholamban, decreasing the rate of Ca^2+^ removal from the cytosol and (2) troponin I, decreasing the actin and myosin bridges detachment rate. This results in an increase in the passive tension of cardiomyocytes (F_passive_). In LV biopsies, F_passive_ was significantly higher in patients with HFpEF compared to patients with HFrEF (Figure 3) [8].

PKGI is involved in maintaining high myocardial compliance through the phosphorylation of the spring elements of titin molecules. At low PKG activity, molecules of the N2B titin isoform are hypophosphorylated, which leads to an increase in F_passive_ and, accordingly, a steeper rise in LV filling pressure [97]. It has been shown that titin hypophosphorylation is more pronounced in HFpEF than in HFrEF [98]. The high stiffness of cardiomyocytes, together with myocardial fibrosis, plays the most important role in myocardial stiffening and increased LV filling pressure [99]. Thus, the enhancement of the NO-cGMP-PKG pathway may be a promising target in HFpEF. In preclinical studies, therapeutic manipulations that target this pathway have improved LV compliance, and reduced cardiac hypertrophy and fibrosis [100,101,102,103]. However, clinical trials with therapeutic interventions targeting the NO-cGMP-PKG pathway by angiotensin receptor neprilysin inhibitors, phosphodiesterase-5 inhibitors, sGC stimulators, NO-inducing drugs including β3 adrenergic receptor agonists, inorganic nitrates, and inorganic nitrates/nitrites showed neutral outcomes in patients with HFpEF [87]. One can assume that a more “natural” activation of this signaling axis and, consequently, improvement of myocardial function, can be achieved via reducing inflammation, oxidative stress, and endothelial dysfunction, namely, via eliminating the main contributors to low NO bioavailability. This approach opens great perspectives for statins, as they would be expected not only to activate the NO-cGMP-PKG pathway but also to reduce diverse pro-inflammatory and oxidative signaling pathways involved in the pathogenesis of HFpEF.

Statins rapidly improve endothelial redox balance and increase the bioavailability of endothelium-derived NO to neighboring cardiomyocytes, macrophages, fibroblasts, etc., leading to the prevention of cardiac hypertrophy and fibrosis and the amelioration of LV diastolic dysfunction (LVDD) [18,104,105]. In a retrospective analysis of biopsy specimens from the study of van Heerebeek et al. [8], patients with HFpEF taking statins had higher protein kinase G activity along with lower cardiomyocyte hypertrophy, lower nitrotyrosine levels (a marker of oxidative processes activity), and lower F_passive_ compared to patients not taking statins [2].

## 6. Effects of Statins on Myocardial Hypertrophy and Fibrosis

LV hypertrophy (LVH) is a common structural cardiac HFpEF alteration [106]. LVH is a compensatory adaptative mechanism against increased demands on LV function, including pressure overload. Subsequently, after a series of poorly characterized events (“transition to failure”), the LV predominantly develops progressive LVDD and elevated filling pressures with preserved EF (HFpEF-phenotype) [107]. LVH is invariably accompanied by LVDD due to delayed relaxation, hypertrophy, increased stiffness of cardiomyocytes, and myocardial fibrosis [52]. Myocardial fibrosis, as well as LVH, apparently represents a compensatory response in the initial stages, providing temporal and spatial coordination of force efforts of both hypertrophied muscle fibers and the whole LV under pathological conditions (high afterload). Thereafter, myocardial fibrosis contributes to increased LV stiffness and LV filling pressures.

The GTP-binding proteins Ras, Rho, and Rac play an important role in LVH [9], and by inhibiting their activity, statins may reduce myocyte hypertrophy [108,109,110]. LVH may be mediated by increased NADPH-oxidase activity and myocardial oxidative stress, while Rac1 (Ras-related C3 botulinum toxin substrate) can directly activate NADPH-oxidase. Statins, via the inhibition of isoprenylation, can suppress Rac1 activation and thereby prevent cardiomyocyte hypertrophy. Mice with a cardiac-specific deletion of Rac1 exhibited reduced NADPH oxidase activity and myocardial oxidative stress, which was accompanied by decreased LVH [111]. In vitro, statins increased the levels of the small GTP-binding protein GDP dissociation stimulator (SmgGDS) and decreased the levels of Rac-1 [112]. In wild-type mice, statins reduced Rac1 expression, myocardial hypertrophy, and fibrosis, but not in mice lacking SmgGDS [113]. In a clinical study, perioperative administration of atorvastatin 40 mg daily to patients undergoing cardiac surgery was accompanied by Rac1-mediated inhibition of NADPH oxidase activity in specimens of the right atrial appendage [105]. In a clinical study of Lee et al. (2002), pravastatin therapy in patients with hypercholesterolemia was accompanied by a decrease in LV mass, presumably through the suppression of free-radical oxidation processes, since in multivariate analysis, a regression of LV mass was correlated only with a change in the level of oxidative damage marker 8-iso-prostaglandin F [110]. In a clinical study by Su et al. (2000), the addition of pravastatin to antihypertensive therapy in hypertensive patients with hypercholesterolemia was associated with an additional decrease in LV mass index regardless of the effect on cholesterol levels [114].

The effect of statins on the myocardium may also be mediated through the inhibition of the Rho/ROCK pathway [23]. The suppression of ROCK activation can lead to numerous beneficial effects, including reduced cell proliferation, myocardial hypertrophy, and fibrosis [9]. Mice with a genetic deletion of ROCK1 were characterized by a lower degree of ischemia-induced myocardial fibrosis compared to wild-type mice [115]. In an experiment with angiotensin II infusion and transverse aortic constriction, mice without ROCK2 had less severe LVH and fibrosis compared with wild-type mice [116]. In another experimental study, mice with a deletion of both ROCK1 and ROCK2 exhibited less severe fibrosis and increased autophagy, compared to mice with a single deletion [117]. Patients with hypertensive LVH had several times higher levels of ROCKs in leukocytes than patients with arterial hypertension but without LVH [118].

Statins may also reduce myocardial fibrosis, presumably via the inhibition of Rho proteins and increased NO bioavailability [119]. Statins were superior to other drugs blocking the activity of the mevalonate pathway (alendronate and fasudil) in their effects on the proliferation of myocardial fibroblasts and inhibited collagen synthesis in vitro [120]. In vitro studies have shown the ability of statins to prevent the differentiation of fibroblasts into myofibroblasts [121] and to induce the reverse differentiation of profibrotic myofibroblasts through the inhibition of GGPP [122], which abrogates a key process in myocardial fibrosis. Simvastatin in vivo suppressed the differentiation of fibroblasts into myofibroblasts and inhibited the angiotensin II-induced cardiac fibrosis by regulating exosome-mediated cell–cell communication within the myocardium [123].

In rat and human cardiac fibroblast cell cultures, atorvastatin significantly decreased collagen synthesis, α(I)-procollagen mRNA expression, and connective tissue growth factor gene expression [124]. In a rat model of LV hypertrophy and HFpEF (Dahl salt-sensitive rats fed with high-salt food), treatment with pitavastatin resulted in decreased expression of hypertrophic and profibrotic genes, which was accompanied by a significant reduction of myocardial fibrosis [125]. In another rat model of HFpEF (transgenic rats overexpressing the renin gene with elevated cardiac levels of angiotensin II), rosuvastatin significantly reduced Rac1 expression, cardiac hypertrophy, and perivascular fibrosis, and improved LV contractility [126]. In a mouse model of HFpEF (C57BL/6 mice with a high-fat diet and N-arginine methyl ester treatment), simvastatin reduced LVDD through a reduction of myocardial fibrosis, which was associated with the prevention of the phosphorylation of Smad (Smad2 and Smad3) and Ras/Raf/MAPK (mitogen-activated protein kinase) profibrotic signaling pathways downstream of the TGF-β receptor in cardiac tissue [127]. In another mouse model of HF (with transverse aortic constriction), simvastatin reduced myocardial fibrosis and hypertrophy by suppressing fibroblast proliferation and myofibroblast formation through modulation of endothelial Krüppel-like factor 2 (Klf2)-TGFβ1 or KLF2-forkhead box P1 (Foxp1, endothelial transcription factor)-TGFβ1 signaling pathways [128]. Lipophilic but not hydrophilic statins increased the expression of cardioprotective interleukin-33 in human cardiomyocytes and cardiac fibroblasts in vitro, presumably through the inhibition of GGPP, which may also contribute to the antihypertrophic and antifibrotic effects of statins [129].

## 7. Effects of Statins on Diastolic Dysfunction and Epicardial Adipose Tissue

Ultimately, the above-mentioned pleiotropic effects of statins may lead to an improvement in LVDD via an increase in myocardial relaxation, F_passive_ reduction, and the mitigation of myocardial fibrosis, which ultimately leads to a decrease in LV filling pressure. The ability of statins to improve LVDD has been shown in multiple preclinical studies [18,130,131,132]. The proposed cellular mechanisms of the beneficial impact of statins on LVDD are presented in Figure 2 and Figure 3.

Clinically, the ability of statins to improve LVDD has also been shown in patients with a variety of cardiovascular conditions, such as ischemic heart disease, hyperlipidemia, and hypertensive heart disease, as well as in patients on peritoneal dialysis. The retrospective cohort study by Okura et al. (2007) enrolled 430 patients with ischemic heart disease but without HF [133]. Statin treatment (31% of participants) was associated with a significant decrease in the ratio of the early mitral flow velocity to early diastolic velocity of the mitral annulus (E/e’ ratio, as a measure of LV filling pressures) and a higher free survival rate (cardiac death and congestive heart failure) compared to patients without statins. In the prospective study by Yagi S. et al. (2011), a low-dose pitavastatin (1.0 mg/day) significantly decreased E/e’ ratio and albuminuria in patients with hyperlipidemia, which was associated with the suppression of oxidative stress [134]. In a retrospective cohort study by Ovchinnikov et al. (2022), involving 223 asymptomatic patients with hypertensive concentric LV hypertrophy and ejection fraction ≥50%, statin treatment significantly prevented the transition to HFpEF and LVDD deterioration during a follow-up of 8 years [135]. In the study by Wu et al. (2017), with 213 patients undergoing peritoneal dialysis and without hyperlipidemia but with elevated hsCRP, atorvastatin significantly improved the tissue Doppler imaging or global strain imaging diastolic parameters [136].

Systemic inflammation, being an important pathologic mechanism of HFpEF, is closely related to the expansion and inflammation of epicardial adipose tissue (EAT). In inflamed EAT, adipocytes and activated macrophages produce proinflammatory cytokines and chemokines, which, via endocrine and paracrine mechanisms, cause local inflammation, oxidative stress, microvascular endothelial dysfunction, and interstitial matrix deposition, thereby leading to slower relaxation, increased cardiomyocyte stiffness, and myocardial fibrosis—the main contributors of LVDD [137]. EAT expansion imposes pericardial restrain and impedes ventricular filling. A meta-analysis of 11 studies showed that increased EAT is independently associated with LVDD [138]. EAT expansion correlates with a poor prognosis in patients with HFpEF [139]. Therefore, the inhibition of EAT expansion and inflammation may be a promising therapeutic strategy for HFpEF.

Statins as anti-inflammatory drugs have some therapeutic potential to attenuate EAT expansion. In patients with aortic stenosis, statin therapy was accompanied by a significant reduction in EAT thickness and levels of EAT-secreted inflammatory mediators [140]. In patients who underwent percutaneous coronary intervention, atorvastatin reduced EAT thickness more significantly than simvastatin in combination with ezetimibe [141]. In patients who underwent pulmonary vein isolation for atrial fibrillation, intensive therapy with atorvastatin (80 mg/day) was accompanied by a significant reduction in EAT thickness [142]. In postmenopausal women with hyperlipidemia, intensive therapy with atorvastatin (80 mg/day) resulted in a greater reduction in EAT, compared to intensive therapy with pravastatin (40 mg/day) [143].

## 8. Statins in Heart Failure

Given the numerous pleiotropic effects of statins, it was once hypothesized that statins might be effective in HF [144]. Data on the use of statins in patients with HFrEF are controversial [145,146,147,148]. In two large randomized, double-blind, placebo-controlled trials (GISSI-HF and CORONA), statins did not improve the prognosis of patients with HFrEF [145,146]. Both studies used hydrophilic rosuvastatin with less pleiotropic potential and low myocardial penetration [149] at a low starting dose of 10 mg, which may have limited its prognostic impact. More recent observational prospective cohort studies and meta-analyses have shown a reduction in mortality following statin treatment in patients with HFrEF [150,151]. Nevertheless, the challenge associated with the use of statins still exists. Current guidelines do not recommend routine use of statins in most patients with HFrEF unless otherwise indicated (hypercholesterolemia, coronary artery disease), but they can be continued in patients who have already been receiving treatment [152].

In contrast, the benefits of statins were consistently observed in HFpEF (see Table 1). The differences may be related to the different pathophysiological drivers for LV remodeling [2,153], suggesting different targets for statins. In an observational study, patients with HFpEF treated with statins were less likely to experience atrial fibrillation [154]. In a prospective study, involving 59 statin-naïve patients with HFpEF, statin therapy consistently reduced blood levels of inflammatory biomarkers (hsCRP and MCP-1), and a biomarker of LV wall stress N-terminal pro-B-type natriuretic peptide (NT-proBNP (NT-proBNP), which was associated with improvement in LVDD and reduction in LV filling pressures [155]. One of the most frequent complications of HFpEF is pulmonary hypertension, occurring in the majority of patients with HFpEF and associated with a poor prognosis [156]. In a retrospective study involving 762 patients with severe pulmonary hypertension (pulmonary artery systolic pressure ≥ 60 mm Hg) and preserved EF (≥50%), the administration of statins was associated with a 58% reduction in overall mortality (*p* < 0.001) and was independent of the presence of chronic obstructive pulmonary disease [157].

Although statin therapy for HFpEF has not been evaluated in randomized trials, many observational studies have consistently reported a positive effect of statins on mortality in HFpEF patients of different ethnicities (Table 1) [13,158,159,160,161,162,163,164,165,166,167,168], including HFpEF patients without coronary artery disease [169]. Several meta-analyses have confirmed the beneficial prognostic effect of statins in HFpEF, where statins reduced all-cause mortality in patients with HFpEF by 25% to 40% [151,170,171]. However, in any case, the results of all these observational studies should be validated in specially designed large randomized clinical trials with sufficient statistical power.

The effects of statins extend beyond HFpEF patients. A higher prevalence of non-cardiac pro-inflammatory and metabolic comorbidities (obesity, hypertension, diabetes, chronic kidney disease, chronic obstructive pulmonary disease, liver disease) in HFpEF may also be relevant [172]. These comorbidities may be involved in the pathophysiology of HFpEF [2]. The association between these comorbidities and myocardial dysfunction may be mediated by systemic inflammation [173], and statins have been shown to be beneficial in these conditions as well [174,175,176].

## 9. Which Statins Are Better for HFpEF?

The ability of statins to pass through the cell membrane depends on their chemical properties and the expression of the appropriate membrane transporters in the tissues [20]. Statins can be divided into hydrophilic (rosuvastatin and pravastatin) and lipophilic (simvastatin, pitavastatin, and atorvastatin), depending on their ability to dissolve in water or lipid-containing media, respectively (Figure 4) [149]. Hydrophilic statins may exhibit greater hepatoselectivity than lipophilic statins, whereas lipophilic statins may undergo passive diffusion through both hepatic and non-hepatic tissues [177]. Active transport of statins into cells is mainly performed by proteins of the OATP (organic-anion-transporting polypeptide) and NTCP (Na^+^ tauro-cholate co-transporting polypeptide) families [177]. Hepatocytes express all types of transporters from these protein families; some members of the OATP family are also found in the membranes of enterocytes, and endothelial and skeletal muscle cells [20]. In contrast, all other cells including myocardial cells—cardiomyocytes, fibroblasts, and macrophages—lack these transporters, so lipophilic but not hydrophilic statins can enter these cells by passive diffusion through cell membranes [177,178]. Hydrophilic statins have a low uptake by cardiac muscles (approximately 1% of liver uptake), whereas lipophilic statins have a higher uptake (up to 80% of liver uptake) [178]. Thus, hydrophilic statins may not exhibit pleotropic effects to the same extent as lipophilic statins, which penetrate extrahepatic tissues such as hypertrophic myocardium.

A meta-analysis of 15 randomized controlled trials showed that lipophilic statins have more pronounced pleiotropic effects compared with hydrophilic statins [179]. Several meta-analyses have shown that lipophilic statins may be more effective in improving cardiac function and reducing inflammation [75,180], as well as reducing all-cause mortality and hospitalizations due to the worsening of heart failure in patients with HFpEF [151,180]. In the Stat-LVDF study, treatment with lipophilic pitavastatin but not hydrophilic rosuvastatin delayed an increase in BNP in patients with hypercholesterolemia and asymptomatic LVDD [181]. Equivalent doses of lipophilic atorvastatin resulted in greater reductions in lipid oxidation markers compared to hydrophilic pravastatin in subjects with hyperlipidemia and metabolic syndrome [182]. In a prospective clinical study by Ovchinnikov et al. (2019), the administration of lipophilic atorvastatin was associated with a more pronounced increase in exercise tolerance, hemodynamic improvement, and a decrease in fibrosis biomarkers compared to hydrophilic rosuvastatin [155].

## 10. Potential Adverse Effects of Statins in HFpEF

Despite the numerous beneficial effects of statins in LVDD/HFpEF, there are known concerns about the adverse effects of statins that may limit their use in HFpEF, particularly muscle myopathy and new-onset type 2 diabetes mellitus (DM) [183].

Myopathy is a frequent dose-dependent side effect of statins, and its occurrence in patients with HFpEF may lead to even greater exercise limitations. Statin-associated muscle symptoms may affect from 5% to 10% of patients. Risk factors for statin-induced myopathy include advanced age, female sex, and physical disability [184], making this a highly relevant issue in HFpEF. In most cases, myopathy manifestations resolve after the discontinuation or reduction of the dose of statins; however, in some patients, this does not happen, which may indicate necrotizing autoimmune myopathy induced by statins [184]. Statins through prenylation inhibition can reduce the expression of ubiquinone (coenzyme Q10), a cofactor in the mitochondrial electron transport chain that can lead to impaired oxidative phosphorylation and energy cell supply, damage, and apoptosis of skeletal muscle cells [183]. In several studies, adding coenzyme Q10 reduced muscle symptoms associated with statins [185]. Statins also potentiate the uncoupling of the stabilizing protein FKBP12 from ryanodine receptor 1, leading to the leakage of Ca^2+^ from the sarcoplasmic reticulum and excessive muscle contraction, thereby provoking myopathy [186]. Impaired activation of small GTPases and the inhibition of the PI3K/Akt pathway may also be important, inducing muscle atrophy via activation of the forkhead box class O (FOXO) protein group followed by an increased expression of atrogenic proteins such as atrogin-1 and muscle-specific ring finger (MuRF)-1 [184].

Statins may increase the risk of DM in a dose-dependent manner by 9% [187], which may be relevant in HFpEF, as DM plays an important role in the pathogenesis of HFpEF. It is believed that these effects of statins are realized through the dysfunction of pancreatic β-cells and insulin resistance of peripheral tissues [188]. Statins can inhibit the expression of the GLUT-2 transporter and inhibit pancreatic ß-cell calcium channels, thereby impairing insulin release [189]. In addition, intensive therapy with statins may induce ß-cell apoptosis [190]. Statins may also reduce peripheral tissue insulin sensitivity via decreased translocation of the GLUT4 transporter and the inhibition of insulin receptor substrate (IRS-1) and its downstream signaling pathways [191]. Importantly, pitavastatin and pravastatin do not affect glucose levels [188].

One possible side effect of statins may be polyneuropathy, especially with long-term exposure. However, according to two meta-analyses and a systematic review, there is no significant association between the use of statins and the risk of polyneuropathy in both non-diabetic and diabetic patients [192].

## 11. Conclusions and Future Perspectives

Currently, the treatment approaches for HFpEF are very limited, and a search for effective treatment is vital. One potential treatment option is statins, which have shown multiple pleiotropic effects on different cell types that may be extremely beneficial in HFpEF (Figure 1). Various hypotheses have been proposed to explain the beneficial effects of statins in HFpEF; however, the exact mechanisms remain incompletely understood and should be confirmed and more rigorously investigated in additional specially designed studies. To date, the ability of statins to improve outcomes in HFpEF has been consistently shown in multiple observational studies, where statins reduced total mortality by 20% to 80%. Importantly, these impressive results were obtained in observational studies, where adherence to therapy is usually significantly lower than in controlled large-scale randomized trials. Thus, these findings should be tested in adequately powered randomized clinical trials. However, the implementation of such studies is severely limited by the fact that statins are indicated for most patients with HFpEF due to a high or very high risk of cardiovascular complications.

Lipid-lowering therapy continues to be a challenge in the treatment of cardiovascular diseases, including elderly patients with HFpEF, as statins are a group of drugs that are most commonly underused and underdosed [193]. In recent large randomized clinical trials (IMPERIOR-PRESERVED, PARAGON-HF, TOP-CAT), 31–47% of participants did not receive lipid-lowering medications [194]. Low adherence to statin treatment in elderly patients is largely due to the paradox of epidemiologic studies showing that low cholesterol in the elderly may be associated with an increased risk of mortality [195]. However, Mendelian randomization studies have shown that high LDL cholesterol levels retain their risk in the elderly as well [196]. The use of statins in elderly patients has been reported to have no additional adverse effects [197]. Regarding the efficacy of statins in older patients with HFpEF, several observational studies have shown that statin use in patients with HFpEF > 75–80 is associated with comparable improved survival compared with younger patients [13,160,165,166]. One of the best ways to improve adherence to statin therapy is counseling and education of patients, combined with frequent and comprehensive monitoring of their adherence.

Since subclinical microvascular inflammation is a common pathogenetic mechanism in HFpEF regardless of the predominant HFpEF phenotype and comorbidities [4], various anti-inflammatory targeting strategies other than statins are being actively tested. An anti-interleukin-1 strategy has held great promise, but clinical trials with a recombinant IL-1 receptor antagonist anakinra in HFpEF have yielded conflicting results [198,199]. To date, it is not clear whether canakinumab, which blocks the effects of IL-1β and has been shown to be highly effective in postinfarction patients [200], can also improve prognosis in HFpEF. Other emerging anti-inflammatory strategies with potential benefits in HFpEF include the administration of colchicine (ongoing COLpEF trial [NCT04857931]), immunomodulation by intracoronary injection of cardiosphere-derived cells (ongoing REGRESS-HFpEF trial [NCT02941705]), myeloperoxidase inhibition (ongoing ENDEAVOR trial [NCT04986202]), and others. A deeper understanding and detailed characterization of the inflammatory mechanisms responsible for the onset and progression of HFpEF is needed to enable more targeted treatment and prognostic improvement in HFpEF.

## Figures and Tables

**Figure 1 ijms-25-04958-f001:**
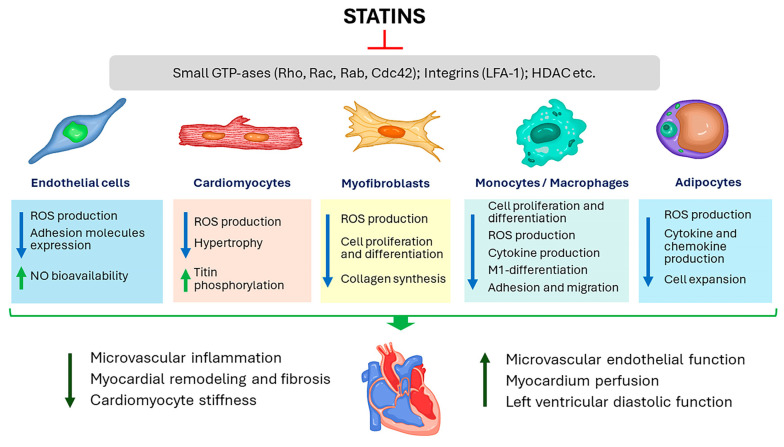
The beneficial myocardial effects of statins in heart failure with preserved ejection fraction (HFpEF) may be due to their impact on different cell types. HDAC indicates histone deacetylase; LFA-1, leukocyte function-associated antigen-1; NO, nitic oxide; ROS, reactive oxygen species. The upward-pointing arrows denote stimulatory and the downward-pointing arrows inhibitory effects.

**Figure 2 ijms-25-04958-f002:**
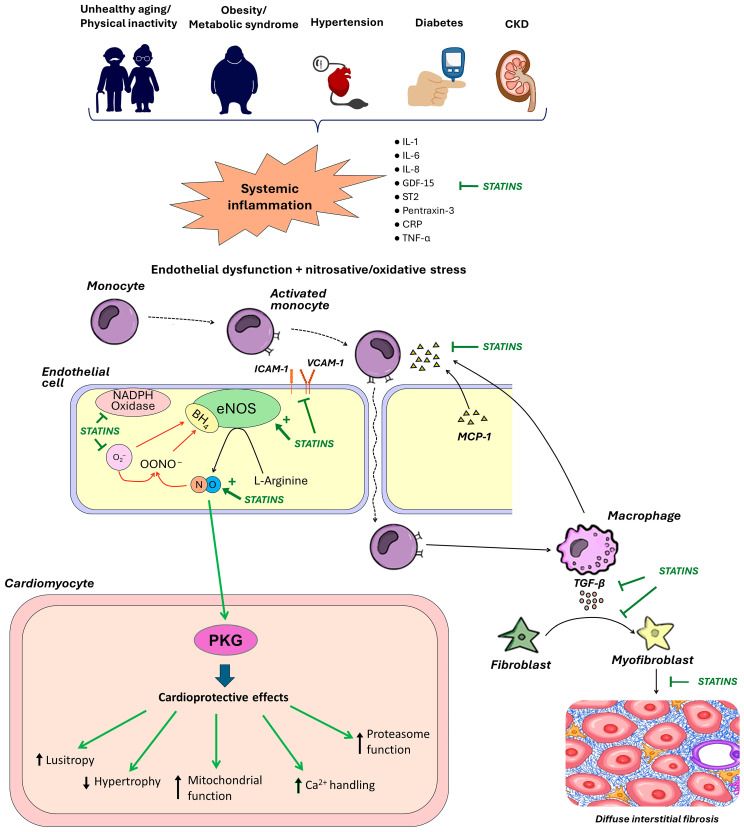
The inflammation paradigm in heart failure with preserved ejection fraction (HFpEF) and potential effects of statins. Unhealthy aging and common pro-inflammatory comorbidities via elevated blood levels of pro-inflammatory cytokines drive chronic systemic low-intensity inflammation leading to coronary microvascular endothelial dysfunction/activation and nitrosative/oxidative stress. Activated/dysfunctional endothelial cells express adhesion molecules (ICAM-1 and VCAM-1) and secrete chemokines (MCP-1), leading to the migration of monocytes from the bloodstream into the myocardium, where they transform into macrophages and release transforming growth factor-β (TGF-β), which promotes the conversion of fibroblasts to myofibroblasts with subsequent deposition of collagen. The formation of reactive oxygen species (ROS) within endothelial cells leads to decreased nitric oxide (NO) bioavailability with subsequent suppression of NO—protein kinase G (PKG) signaling and cardiac structural and functional adverse changes. Statins have numerous pleiotropic effects and may be useful at all stages of this inflammation paradigm. Statins reduce the production of proinflammatory cytokines and decrease the expression of adhesion molecules and chemokines. In endothelial cells, statins induce endothelial NO synthase (eNOS) and reduce ROS levels through direct ROS scavenging and suppression of NADPH-oxidase activity. This prevents eNOS uncoupling by suppressing the oxidation of a cofactor of eNOS tetrahydrobiopterin (BH4) via peroxynitrite (OONO^−^), which is formed by the interaction of superoxide anions (O_2_^−^) and NO. Ultimately, all these effects increase NO bioavailability and restore the NO-PKG signaling network with numerous resultant beneficial effects. Statins may also reduce myocardial fibrosis, presumably via the prevention of fibroblast differentiation, the inhibition of profibrotic signaling pathways downstream of the TGF-β receptors, and collagen synthesis. CKD, chronic kidney disease; GDF-15, growth differentiation factor-15; hsCRP, high-sensitivity C-reactive protein; ICAM-1, intercellular adhesion molecule 1; IL, interleukin; MCP-1, monocyte chemoattractant protein-1; NADPH, nicotinamide adenine dinucleotide phosphate; TNF-α, tumor necrosis factor-α; VCAM 1, vascular cell adhesion molecule 1; ST2, soluble suppression of tumorigenicity 2. The arrow lines denote stimulatory and ☐ lines inhibitory effects.

**Figure 3 ijms-25-04958-f003:**
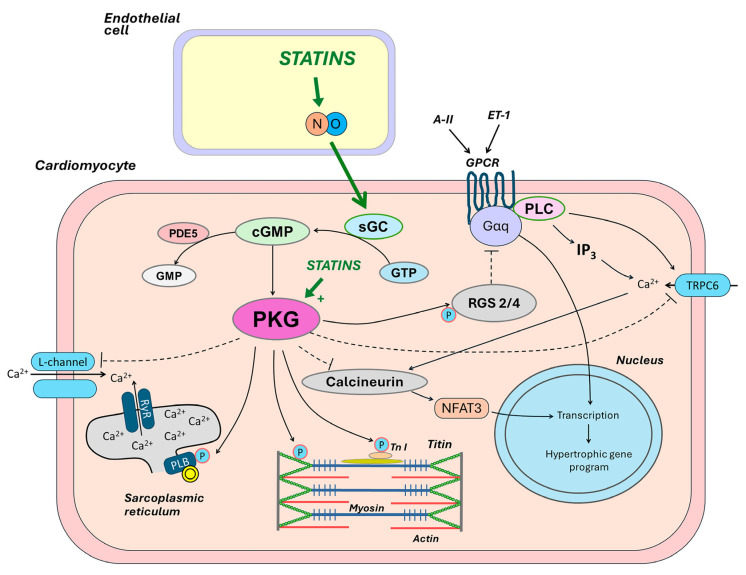
The role of statins in increasing NO-cGMP-PKG pathway activity and improving myocardial function. Statins increase the bioavailability of endothelium-derived NO to neighboring cardiomyocytes and restore the NO-cGMP-PKG signaling network with resultant numerous beneficial effects. PKG mediates phosphorylation (P) of phospholamban (PLB) to activate sarcoplasmic reticulum (SR) calcium ions (Ca^2+^)-ATPase pump and increase Ca^2+^ uptake into the SR, as well as the phosphorylation of troponin I (Tn I) to reduce myofilament Ca^2+^ sensitivity and thereby increase lusitropy. The PKG-mediated phosphorylation of titin reduces cardiomyocyte stiffness. PKG suppresses prohypertrophic signaling cascades, including the Gq-coupled pathway via the phosphorylation of the regulator of G protein signaling (RGS) 2/4, and the calcineurin/ nuclear factor of activated T-cells (NFAT) pathway via the phosphorylation of the canonical transient receptor potential Ca^2+^ channel (TRPC) 6 signaling. A-II indicates angiotensin-II; ET, endothelin; GPCR, G-protein controlled receptor; GTP indicates guanosine triphosphate; IP3, inositol triphosphate; PDE5, phosphodiesterase-5; RyR, ryanodine receptor; PLC, phospholipase C; sGC, soluble cyclase; Tm, tropomyosin.

**Figure 4 ijms-25-04958-f004:**
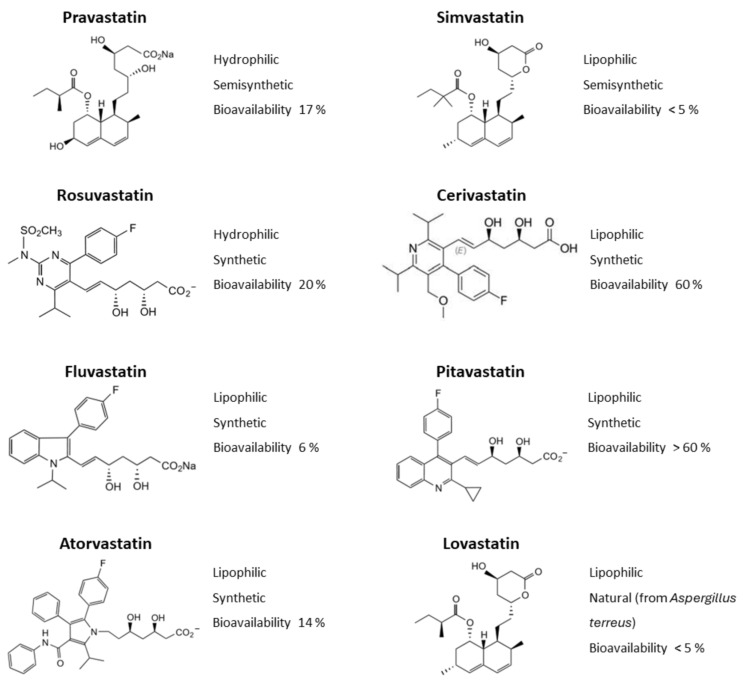
Statin structure and properties.

**Table 1 ijms-25-04958-t001:** A summary of clinical studies evaluating the effect of statins on prognosis in patients with heart failure with preserved ejection fraction.

Study, Year [Reference]	Study Design	Country	Patients, n	Proportion on Statins, %	Type of Statin	Follow-Up, Years	Main Results
Fukuta H. et al., 2005 [158]	Prospective	United States	137	50	Atorvastatin, simvastatin, pravastatin (Lp/Lp/Hp)	2	↓ All-cause mortality (HR 0.20; 95% CI 0.06 to 0.62; *p* < 0.01)
Roik M. et al., 2008 [159]	Prospective	Poland	146	71	Simvastatin, atorvastatin, lovastatin (Lp/Lp/Lp)	1	↓ All-cause mortality (HR 0.24, 95% CI 0.07 to 0.90; *p* < 0.05) ↓ CV admission (HR 0.55, 95% CI 0.33 to 0.92; *p* < 0.05)
Shah R. et al., 2008 [160]	Retrospective	United States	13,533	17	N/A	3	↓ All-cause mortality (HR 0.73, 95% CI 0.68 to 0.79, *p* < 0.001)
Tehrani F. et al., 2010 [161]	Retrospective	United States	270	30	N/A	5	↓ All-cause mortality (HR 0.65; 95% CI 0.45 to 0.95; *p* = 0.029) No significant reduction in CV admission
Gomez-Soto F.M. et al., 2010 [162]	Prospective	Spain	1120	50	Simvastatin, lovastatin, pravastatin (Lp/Lp/Hp)	5	↓ All-cause mortality (HR 0.34, 95% CI 0.21 to 0.47, *p* < 0.001) ↓ HF admission (13.9 vs. 19.7 per 100 persons-year, *p* < 0.001)
Quirós López R. et al., 2012 [163]	Retrospective	Spain	231	21	N/A	10	↓ All-cause mortality (HR 0.27, 95% CI 0.15 to 0.50, *p* = 0.01)
Kaneko H. et al., 2013 [164]	Prospective	Japan	1121	55	N/A	3	↓ All-cause mortality (HR 0.26, 95% CI 0.140 to 0.479, *p* < 0.001)
Nochioka K. et al., 2015 [165]	Prospective	Japan	3124	37	N/A	3	↓ All-cause mortality (HR 0.71; 95% CI 0.62 to 0.82; *p* < 0.001.) No significant reduction in HF admission
Alehagen U. et al., 2015 [166]	Prospective	Sweden	9140	38	N/A	1	↓ All-cause mortality (HR 0.80, 95% CI 0.72 to 0.89, *p* < 0.001) ↓ All-cause mortality + CV admission (HR 0.89, 95% CI 0.82 to 0.96; *p* < 0.01)
Yap J. et al., 2015 [167]	Prospective	Singapore	751	61	N/A	2	↓ All-cause mortality (HR 0.59, 95% CI 0.44 to 0.79; *p* < 0.001) ↓ CV mortality (HR 0.58, 95% CI 0.37–0.89; *p* = 0.012)
Lee M.S.et al., 2018 [13]	Retrospective	United States	7563	72	N/A	6.7	↓ All-cause mortality (HR 0.73, 95% CI 0.66 to 0.81; *p* < 0.001)
Tsujimoto T. et al., 2018 [168]	Prospective (data of TOPCAT trial)	United States, Canada, Brazil, Argentina, Russia, Georgia	3378	52	N/A	3.3	↓ All-cause mortality (HR 0.79, 95% CI 0.63 to 0.99; *p* = 0.04)
Marume K. et al., 2019 [169]	Prospective	Japan	414	20	N/A	2	↓ All-cause mortality (HR 0.21, 95% CI 0.06 to 0.72; *p* < 0.02)

CI, confidence interval; CV, cardiovascular; Hp, hydrophilic; HR, hazard ratio; Lp, lipophilic; ↓, decrease.

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
