# Peer review of "Use of Statins in Heart Failure with Preserved Ejection Fraction: Current Evidence and Perspectives"

_ijms, 2024, doi:10.3390/ijms25094958_

Round 1

Reviewer 1 Report

Comments and Suggestions for Authors

The author effectively conveyed the content of the manuscript and informative for understanding the statins therapy for the heart failure.  Clarify the unique contribution of the manuscript, considering that several review articles on the same topic have already been published? 

1.     Discuss the constraints of statins in the context of heart failure? Can the use of statins potentially cause polyneuropathy and what is their impact on cholesterol levels? And how it contributes to further cardiovascular diseases.

2.   Explain how different age groups are affected by statins in relation to heart failure? And their clinical importance with respective age groups.

3.     Explain the Effect of Statins on Diastolic Dysfunction like “section 7: Effect of Statins on Diastolic Dysfunction and Epicardial Adipose Tissue”?

4.     As mentioned in the manuscript use of statin resulted in a decrease in collagen synthesis. Does lower collagen production have any impact on blood flow/blood vessels and subsequently on heart failure?

5.     The author should utilize the abbreviations mentioned in the manuscript following their initial mention. In line 74 on page 2, the author made reference to Nitric oxide, which was previously referenced using the short form NO.

6.     Thoroughly review the grammar throughout the manuscript.

Reviewer 2 Report

Comments and Suggestions for Authors

 The manuscript “Statins for Heart Failure with Preserved Ejection Fraction: Cur-2 rent Evidence and Perspectives” by Artem Ovchinnikov et al is a very interesting review of a relevant cardiovascular topic. The report is well-written.

While I acknowledge the quality of IJMS, I am confident that this manuscript has the potential to make a significant impact in a journal with a more clinical/cardiovascular approach. I strongly recommend its evaluation and publication on such a platform; MDPI has several journals in this regard.

I have some minor comments.

Line 51-53 “In contrast, in HFpEF, remodeling is initially governed by chronic microvascular inflammation, which is supported by compelling preclinical and  clinical evidence [4].”

Please choose other references since Pugliese et al [4] do not report any preclinical studies.  

Figure 1 & 2 resolution is low. Please increase it

A figure with names and structures of statins will be helpful for the readers

Comments on the Quality of English Language

Minor edition

Reviewer 3 Report

Comments and Suggestions for Authors

The authors should be congratulated on an excellent publication. Please find my queries are listed below:

Abstract

1. Though commonly employed, please define the abbreviation "HF" (line 23) prior to use of the abbreviation. This can easily be accomplished in line 21.

Introduction

2. In reference to line 38, please also acknowledge that increasing diagnostic power/capabilities is a confounder to this increasing prevalence and incidence of HFpEF.

3. Please elaborate on what is meant by "unsatisfactory" (line 40)

Part 4

4. Figure 2 may benefit from being split into two separate figures, which would also allow for the lengthy figure legend to be reduced in volume.

5. In regard to line 270-1, please clarify positive versus negative acute phase reactants, and also clarify that though C-reactive protein is a sensitive marker, its widespread adoptance by clinical laboratories due to its ease of detection is another reason for its success. 

Part 5

6. Please rephrase the section title "Low NO-cGMP-PKG Pathway Activity Contributes to Myocardial Dysfunction. Role of Statins", such that it is contained within one sentence

Part 8

7. Please clarify what is meant by "(see below" in line 518

Part 9

8. This reviewer wonders if Table 1 should describe lipophilic versus liphophobic statins in column six, rather than, or in addition to, naming statins by name

Overall

9. Though this topic continues to undergo refining as additional clinical data become available, this topic is not novel. The article would benefit from an explicit statement on what novelty this article brings to the field, to assist in the justification for publication
